# Upregulated GATA3/miR205-5p Axis Inhibits MFNG Transcription and Reduces the Malignancy of Triple-Negative Breast Cancer

**DOI:** 10.3390/cancers14133057

**Published:** 2022-06-22

**Authors:** Samson Mugisha, Xiaotang Di, Doudou Wen, Yuetao Zhao, Xusheng Wu, Shubing Zhang, Hao Jiang

**Affiliations:** 1Department of Cell Biology, School of Life Sciences, Central South University, Changsha 410013, China; mgsam02@yahoo.fr (S.M.); dixiaotang@csu.edu.cn (X.D.); doudou_wen@csu.edu.cn (D.W.); shubingzhang@csu.edu.cn (S.Z.); 2Department of Biochemistry and Molecular Biology, School of Life Sciences, Central South University, Changsha 410013, China; ytzhao@csu.edu.cn; 3Shenzhen Health Development Research and Data Management Center, Shenzhen 518000, China; szjysgwuxusheng@gmail.com; 4Hunan Key Laboratory of Animal Models for Human Diseases, Central South University, Changsha 410013, China; 5Department of Biomedical Informatics, School of Life Sciences, Central South University, Changsha 410013, China

**Keywords:** breast cancer, TNBC, MFNG, GATA3, miR205-5p, metastasis

## Abstract

**Simple Summary:**

Triple-negative cancer (TNBC) is a deadly disease that presents a potential health threat to women worldwide. It is the most aggressive and presents a poor prognosis among all breast cancer subgroups. We previously demonstrated that the elevated expression of manic fringe (MFNG) plays a pivotal role in breast cancer. However, the mechanism through which MFNG is regulated remains obscure. The study presented here set out to determine the mechanism by which MFNG expression is regulated in TNBC. Our findings revealed that GATA3 and miR-205-p cooperatively block the transcription of MFNG leading to the inhibition of cell migration and tumor growth in vitro and in vivo. Our study uncovers a novel GATA3/miR-205-p/MFNG feed-forward loop and miR205-5p could be adopted as a potential therapeutic strategy of TNBC.

**Abstract:**

Triple-negative breast cancer (TNBC) accounts for approximately 20% of all breast carcinomas and has the worst prognosis of all breast cancer subtypes due to the lack of an effective target. Therefore, understanding the molecular mechanism underpinning TNBC progression could explore a new target for therapy. While the Notch pathway is critical in the development process, its dysregulation leads to TNBC initiation. Previously, we found that manic fringe (MFNG) activates the Notch signaling and induces breast cancer progression. However, the underlying molecular mechanism of MFNG upstream remains unknown. In this study, we explore the regulatory mechanisms of MFNG in TNBC. We show that the increased expression of MFNG in TNBC is associated with poor clinical prognosis and significantly promotes cell growth and migration, as well as Notch signaling activation. The mechanistic studies reveal that MFNG is a direct target of GATA3 and miR205-5p and demonstrate that GATA3 and miR205-5p overexpression attenuate MFNG oncogenic effects, while GATA3 knockdown mimics MFNG phenotype to promote TNBC progression. Moreover, we illustrate that GATA3 is required for miR205-5p activation to inhibit MFNG transcription by binding to the 3′ UTR region of its mRNA, which forms the GATA3/miR205-5p/MFNG feed-forward loop. Additionally, our in vivo data show that the miR205-5p mimic combined with polyetherimide-black phosphorus (PEI-BP) nanoparticle remarkably inhibits the growth of TNBC-derived tumors which lack GATA3 expression. Collectively, our study uncovers a novel GATA3/miR205-5p/MFNG feed-forward loop as a pathway that could be a potential therapeutic target for TNBC.

## 1. Introduction

Breast cancer is the most commonly diagnosed cancer with ~2.261 million new cases and ~684,000 deaths, making breast cancer the fifth deadliest cancer, according to Global Cancer Statistics 2020 [1]. It is classified into molecular subtypes with different clinical and pathological features, treatment responses, and outcomes [2]. Among them is triple-negative breast cancer (TNBC), which lacks the expression of progesterone (PR), estrogen (ER), and human epidermal growth factor receptor 2 (HER-2) [3]. Compared to other molecular subtypes of breast cancer, TBNC has the poorest survival pattern due to its aggressiveness and higher metastatic potential and chemoresistance [4].

Atomically thin black phosphorus (BP) is a new member of the 2D inorganic materials family. It has been a promising alternative carrier to deliver a drug in cancer therapy because it could passively accumulate in tumors at more significant concentrations via the well-known enhanced permeability and retention effect [5,6]. Hence, BP is emerging as a suitable nanoparticle that allows drug delivery and therapeutics with minimal invasiveness and high therapeutic efficiency [7].

Notch signaling is a high evolutionally conserved pathway. Recent reports have underlined the implication of Notch signaling upregulation in the TNBC. Increased Notch1 stimulates epithelial to mesenchymal transition (EMT) and induces tumor formation in TNBC [8]. In mammals, Notch signaling is controlled by glycosyltransferases known as fringes: lunatic fringe (LFNG), manic fringe (MFNG), and radical fringe (RFNG) through their fucose-specific β-1,3-N-acetylglucosaminyltransferase activity. This process leads to O-linked fucose residues elongation towards the Notch receptor which triggers the activation of the Notch signaling [9,10]. Of the fringes, MFNG acts as a tumor suppressor in lung cancer by inhibiting the activation of Notch3 [11]. Conversely, MFNG is upregulated in kidney clear cell renal carcinoma (KIRC) which could be linked to kidney cancer progression, hence an oncogene [12]. The MFNG controversy underlines its dual function depending on the cancer type. Previously, we reported that MFNG was highly expressed in claudin-low breast cancer (CLBC) and promoted cell migration, proliferation, and tumorigenicity by modulating Notch activation by directly targeting phosphoinositide kinase (PIK3CG) [13]. However, the involvement of MFNG in TNBC progression has not been previously reported and the mechanism in which MFNG is regulated also remains unclear.

GATA3, a zinc-finger transcription factor belonging to the GATA family, has been reported to suppress cancers by regulating their target genes and acts as a master regulator of mammary gland differentiation and homeostasis [14]. GATA3 inhibits prostate cancer progression by upregulating miR-503 expression to repress the oncogenic activity of ZNF217 [15], and abrogates breast cancer metastasis through regulating Semaphorin 3B expression [16]. Based on the recent studies, GATA3 expression is frequently reduced in TNBC cells [17,18]; however, the aberrantly activated signaling which confers TNBC malignancy is still unknown.

Therefore, our study aimed to explore the regulatory network between GATA3 and MFNG in TNBC progression. We found that MFNG was highly expressed in TNBC and the attenuated MFNG expression could reduce the malignancy of TNBC. In the mechanistic study, we demonstrated that MFNG is a direct target of GATA3 and miR205-5p and GATA3 could bind to the miR205-5p promoter to elevate its expression. We revealed that GATA3 and miR205-5p formed a coherent feed-forward loop to cooperatively abrogate MFNG expression. Additionally, the combination of nanoparticles polyetherimide-black phosphorus (PEI-BP) with miR205-5p (PEI-BP-miR205-5p) significantly reduced the tumorigenesis of TNBC-derived tumors. Our study explored a novel feed-forward loop downstream of GATA3 that suppresses the progression of TNBC.

## 2. Materials and Methods

### 2.1. Cells and Tissue Specimens

The human breast cancer cells MCF-7, T47D, BT-549, MDA-MB-231, MDA-MB-468, HCC-1806, and HS578-T were obtained from Zhiyong Luo’s lab (School of Life Sciences, Central South University, Changsha, China). Among these cell lines, BT-549 and MDA-MB-231 were selected for further experiments as they share major characteristics such as belonging to the same TNBC subtype [19]. The cells were grown in DMEM with 10% fetal bovine serum (FBS) and antibiotics 100 U/mL penicillin and 100 µg/mL streptomycin (P/S) and were maintained in an incubator at 37 °C supplied with 5% CO_2_.

After obtaining written informed consent, all tissue specimens were collected from Xiangya Second Hospital, Central South University (Changsha, China). These experiments were approved by the Ethical Committee of the School of Life Sciences, Central South University (Changsha, China) following the Declaration of Helsinki ethical guidelines.

### 2.2. Generation of Stable Cells That Overexpress MFNG and GATA3

GATA3 cDNA was extracted from MDA-MB-231 cells, cloned into a pCMV-neo vector, and transfected into HEK-293T cells using Lipofectamine 2000 (TIANGEN, Beijing, China). Seventy-two hours after transfection, the supernatant was harvested and filtered using 0.22 µm pore size filters. The virus-containing medium was titrated to infect BT-549 and MDA-MB-231 for 48 h. Afterward, the infected cells were selected with 2 µg/mL puromycin for 14 days.

### 2.3. Genomic DNA, RNA Isolation, shRNA, PCR, and RT-qPCR

The genomic DNA from MDA-MB-231 cells was isolated and purified using the Genomic DNA Purification Kit following the manufacturer’s instructions (TIANGEN, Beijing, China). The total RNA was subsequently reversed and transcribed to cDNA in standard conditions. Real-time qPCR analyses were carried out using SuperReal PreMix Plus (SYBR Green) (TIANGEN, Beijing, China) as the supplier required. Data were obtained using the 2^−ΔΔCT^ technique, where ΔCT = CT_gene tested_—CT_GAPDH_, and normalized to GAPDH expression. For each sample, data were generated from triple reactions. Non-targeting control shRNA and shRNAs against human MFNG and GATA3 were purchased from Sangon, Shanghai, China. The shRNAs were then transfected into BT-549 and MDA-MB-231 cells using Lipofectamine 2000 as the manufacturer instructed. The sequences of shRNA and RT-qPCR primers were provided in Appendix A, respectively.

### 2.4. Western Blot Assay

BT-549 and MDA-MB-231 cells were lysed in RIPA buffer (Beyotime Biotechnology, Beijing, China). Lysates were centrifuged at 13,000× *g* for 10 min at 4 °C, and the protein concentration was quantified using a Pierce Bicinchoninic Acid (BCA) Protein Assay Kit (Beyotime Biotechnology, Beijing, China). In total, 25 µg of protein samples was separated in 10% SDS-PAGE, which were then electro-transferred to a polyvinylidene fluoride (PVDF) membrane (Biorad). The membranes were blocked in 5% non-fat dry milk for one hour at room temperature and then incubated overnight with the primary antibody at 4 °C. The membranes were treated with a secondary antibody for 1 h at room temperature after being washed 3 times with TBST. The blots were developed using the Efficient Chemiluminescence Kit (GENVIEW) and SageCapture imaging System (SAGECREATION). Specific proteins were detected with their specific antibodies. Anti-MFNG antibody (NBP1-79288, 1:1000) was purchased from Novus Biologicals (USA), anti-E-Cadherin antibody (24E10, 1:100), anti-Flag antibody (D6W5B, 1:1000), anti-Vimentin antibody (D21H3, 1:1000), anti-SLUG antibody (C19G7, 1:1000), anti-SOX2 antibody (D9B8N, 1:1000), and anti-ß-Catenin antibody (D10A8, 1:1000) were purchased from Cell Signaling Technology (Danvers, MA, USA), anti-GATA3 antibody (HG3-3, 1:1000) was purchased from Santa Cruz Biotechnology, Inc., and anti-GAPDH antibody (1E6D9, 1:10000) was purchased from Proteintech, (Rosemont, IL, USA). All the above were applied as primary antibodies. While anti-rabbit and anti-mouse HRP conjugated (1:4000) were purchased from Beyotime Institute of Biotechnology Inc. (Shanghai, China) and utilized as secondary antibodies.The original western blot images are shown in Appendix A.

### 2.5. Luciferase Reporter Assay

Cells were transfected with lipofectamine 2000 (Invitrogen Life Technologies, Waltham, MA, USA) for 48 h. In brief, the MFNG promoter-reporter was created by subcloning the MFNG promoter (−510 to +100) into the luciferase reporter vector pGL4.22 (Promega, Madison, WI, USA) which contains an SV40 promoter using the restriction enzymes XhoI and EcoRV. http://jaspardev.genereg.net/ (accessed on 24 March 2019) and Targetscan and miRDB were used to predict the potential binding sites for the GATA3 and miR205-5p. BT-549 and MDA-MB-231 cells were co-transfected with MFNG promoter-reporter and Renilla luciferase plasmids. Moreover, we designed and cloned the wild-type MFNG 3′-UTR sequence (WT) and the mutant into an empty pmirGLO luciferase vector (Promega). MDA-MB 231 cells were seeded into a 24-well plate and transfected with the WT, mutant, and 50 nmol/L miR205-5p mimic and their respective negative control (NC). Luciferase reporter assay was performed afterwards using luciferase assay system E1501 kit (Promega) to measure luciferase activities following the manufacturer’s standard instructions.

### 2.6. Gene Expression and Correlation Analysis of the Public Database

The TCGA (nature, 2012) data, including MFNG expression (Z-score) and breast cancer clinical information, were obtained from cBioPortal for Cancer Genomics database. Based on this TCGA data, we analyzed the expression level of MFNG in non-TNBC (all breast cancers present ER, PR, and/or HER-2 positive) and TNBC samples and assessed the correlation between MFNG expression and overall survival in breast cancers, non-TNBC, and TNBC. Survival curves were plotted by the Kaplan–Meier method and analyzed by the log-rank test. The correlation between MFNG and GATA3 expression was investigated by the GEPIA2 database using an online correlation assay.

### 2.7. Cell Growth Assay

CCK-8 kit (TIANGEN, Beijing, China) was employed to determine cell proliferation as per the manufacturer’s instruction. BT-549 and MDA-MB-231 cells were transfected with pLenti-CMV-MFNG-FLAG-GFP-Puro, pLV-CMV-GFP-GATA3, and miR205-5p with their respective controls using Lipofectamine 2000 (TIANGEN). At 48 h post-transfection, 3 × 10^3^ cells/well cells were seeded into 96-well culture plates. The CCK-8 reagent at (1:10) ratio was then added at 0, 24, 48, 72, and 96 h. After 2 h of incubation of cells with CCK-8 reagent, the optical density (OD) was read at 450 nm using the Varioskan Flash Spectral Scanning Multimode Reader (Thermo Fisher Scientific, Waltham, MA, USA).

### 2.8. Transwell Migration and Wound-Healing Assays

After transfection, BT-549 and MDA-MB-231 cells were starved for 12 h and transferred into the transwell migration upper chambers for the cell migration analysis. DMEM supplied with 0.5% FBS in the lower chambers were loaded simultaneously. After 24 h, cells in the upper chambers were removed, and cells that had moved into the lower chambers were fixed with a 4% paraformaldehyde (PFA). The membrane (lower chamber) with migrated cells was then stained with crystal violet and the cells in the upper chamber were wiped with a wet cotton swab. After the chamber was slightly dry, it was placed under the Leica fluorescent microscope (Leica Microsystems GmbH, Wetzlar, Germany) for photographing. The average of each group was measured, and the change in cell migration ability was calculated by the number of cells passed through the membrane. Wound-healing assay was utilized for directional cell migration analysis.

### 2.9. Colony Formation Assay

Transfected BT-549 and MDA-MB-231 cells were collected with 0.25% trypsin, seeded at 1 × 10^3^ cells in each well of a 6-well plate, and grown in serum-free DMEM for 2 to 3 weeks. Cells were then trypsinized, enumerated, and mixed in a DMEM supplied with 0.35% agarose. Three thousand cells per well were seeded on a 0.5% agarose layer in a 6-well plate. Colonies of seeded cells were allowed to develop. Colonies were generated after 14 days and fixed for 20 min in 600 mL of 4% PFA, then stained with crystal violet reagent for 20 min for quantification via counting after photographing.

### 2.10. Immunohistochemistry

Mammary tissues extracted from tumors were fixed overnight in 4% PFD, dehydrated, and paraffin-embedded. Sections (5 µm) were rehydrated, followed by heating for antigen retrieval, and stained with X-gal overnight to observe tissue structure according to standard protocols (Buono et al., 2006), then immunohistochemical analysis by standard procedures was performed. Rabbit MFNG-antibody (NBP1-79288, Novus Biologicals) and Mouse Anti-GATA3 Antibody (HG3-31): sc-268 (Santa Cruz Biotechnology, Co., Ltd., Dallas, TX, USA) were used as primary antibodies and diluted at a 1:100 ratio. Staining was visualized by using HRP/DAB Detection IHC Kit (GK500710, Gene Tech, Shanghai Company Limited, Shanghai, China), sections were counterstained with hematoxylin and photographed using fluorescence and bright-field color imaging microscope (Japan) for further analysis. Each sample’s immunohistochemistry was performed three times. The expression of GATA3 and MFNG in immunohistochemistry sections was evaluated using a German semiquantitative scoring system based on staining intensity and area. The intensity of staining (0: no staining, 1: weak staining, 2: moderate staining, and 3: strong staining) and the percentage of stained cells (0 = 0%, 1 = 1–24%, 2 = 25–49%, 3 = 50–74%, and 4 = 75–100%) were assigned to each specimen. The final immunoreactive score was calculated by multiplying the intensity score by the percentage of stained cells score. Consequently, nine grades were assigned as 0, 1, 2, 3, 4, 6, 8, 9, and 12, and the results were analyzed.

### 2.11. Chromatin Immunoprecipitation (ChIP) Assay

ChIP assay was performed using the ChIP-Grade Protein G Magnetic Beads (Cell Signaling Technology, Inc., Danvers, CO, USA). Briefly, MDA-MB-231 cells were sonicated after treatment with 1% PFA, neutralized, and resuspended in SDS lysis solution (1% SDS, 10 mM EDTA, 50 mM Tris, pH 8.0, and PMSF) for chromatin fragmentation. Sheared chromatin was diluted and normal IgG was used for immunoprecipitation. Reversing the crosslinking and digestion with proteinase K were used to extract DNA from immunoprecipitates, which was subsequently PCR amplified. The ChIP qPCR data were analyzed as previously described [20].

### 2.12. Nanoparticles-miR205-5p Preparation

Briefly, black phosphorus nanoparticles (BP) with a particle size of 30–40 nm were prepared by the liquid-phase exfoliation method and the surface of nano-black-phosphorus was modified with polyacetimide (PEI) to make it positively charged (PEI-BP). The negatively charged miR205-5p or negative control (NC) was loaded on the surface of BP nanoparticles by electrostatic adsorption to construct a miRNA delivery system (miR205-5p-PEI-BP, NC-PEI-BP). The transmission electron microscope and ultraviolet-visible-near-infrared light absorption spectrum confirmed the link between miR205-5p and PEI-BP at a 500–900 nm wavelength.

### 2.13. Tumor Xenograft Model in Mice

All animal care and procedures were carried out in accordance with National Institutes of Health (NIH) guidelines and were approved by the Ethics Committee of the State Key Laboratory of Medical Genetics (NO. 2016030901). The nude mice were purchased and kept in the CSU animal laboratory facility under pathogen-free conditions. Briefly, for the tumor growth experiment, 5 × 10^5^ MDA-MB-231 cells were resuspended in 20 μL DMEM and mixed at a 1:1 ratio with 20 μL Matrigel (BD Bioscience) on ice. In total, 40 μL of the mixture was injected into the mammary fat pad of four-week-old mice. The mice xenograft study involving TNBC cells (MDA-MB-231) included two groups: the first group was injected with MDA-MB-231 cells and treated with miR205-5p-PEI-BP (5 nmol) twice a week, while the second group was the negative control treated with NC-PEI-BP (5 nmol). All groups were made up of 8 to 10 mice. The tumor volume was measured every three days using the formula: tumor volume = length × width × width/2. Finally, after 2 weeks the mice were sacrificed, and tumors were excised for further analysis. Two-way ANOVA analysis was used to assess statistical differences in tumor growth.

### 2.14. Statistical Analysis

GraphPad Prism 7.0 and SPSS17.0 were used for statistical analysis. Survival curves were plotted by the Kaplan–Meier method and analyzed by the log-rank test. Differences between groups were analyzed by Student’s *t*-test and two-way ANOVA analysis. The mean ± standard deviation (SD) of data from at least three different experiments was displayed. The data were judged statistically significant at * *p* < 0.05, ** *p <* 0.01, and *** *p* < 0.001.

## 3. Results

### 3.1. MFNG Expression Was Significantly Increased in TNBC and Positively Associated with Poor Prognosis of TNBC Patients

To evaluate the MFNG expression levels in TNBC and non-TNBC subgroups, we analyzed the clinical data obtained from the cBio Cancer Genomics Portal database (TCGA, nature 2012). The result showed that MFNG was expressed at a higher level in TNBC compared to non-TNBC (*p* = 0.0249) (Figure 1A). We also assessed MFNG expression in TNBC and non-TNBC tissues using IHC staining and similar results were observed (*p* = 0.013) (Figure 1B). To further confirm the above result, we investigated the MFNG expression in breast cancer cells and found that MFNG was highly expressed in TNBC cells (Figure 1C). Moreover, using the same clinical data, we classified patients into MFNG high and low expression groups based on the expression levels. The correlation between MFNG expression and prognosis of TNBC patients was assessed, the result indicated that high expression of MFNG was positively correlated with worse overall survival of TNBC (*p* < 0.001) patients, not significantly correlated with those of breast cancer (*p* = 0.5557) and non-TNBC (*p* = 0.8409) patients (Figure 1D–F. In general, we uncovered that MFNG was highly expressed in TNBC tissues, and higher MFNG expression predicted a poor prognosis for TNBC patients.

### 3.2. MFNG Expression Enhanced the Cell Growth and Migration of TNBC Cells

The above observations prompted us to investigate the function of MFNG in the malignant behavior of TNBC. We overexpressed MFNG and knocked it down using shRNA sequences in TNBC cells and the MFNG expression level was analyzed by RT-qPCR (Figure 2A) and Western blot (Figure 2B). The effect of MFNG on cell growth and migration was examined through colony formation, transwell, and CCK-8 assays. We detected that forced expression of MFNG notably increased the cell viability and migration of TNBC cells. In contrast, the knockdown of MFNG remarkably suppressed the cell migration and growth of TNBC cells (Figure 2C–E). Furthermore, we examined the expression of the factors involved in growth and epithelial to mesenchymal (EMT) in stable cells overexpressing MFNG. The results indicated that the EMT markers Vimentin and SLUG were upregulated, E-cadherin was downregulated, and growth-related marker β-catenin was increased following MFNG overexpression (Figure 2F). It was reported that MFNG could promote the malignancy of breast cancer by activating Notch signaling; therefore, we evaluated whether the ectopic expression of MFNG could activate Notch signaling. Notably, the MFNG upregulated Notch target genes HES1 and HEY1 expression in TNBC cells (Figure 2G). These results collectively suggest that MFNG enhanced cell growth and migration in TNBC by elevating Notch signaling.

### 3.3. Decreased GATA3 Elevated the Malignancy of TNBC Cells and Promoted the Activity of Notch Signaling

It has been previously reported that GATA3 expression is low in TNBC [16] and its deficiency or alterations lead to breast cancer aggressiveness [21]. To identify its functional role in TNBC progression, we established BT-549 and MDA-MB-231 stable cells overexpressing and knockdown GATA3, which was verified by RT-qPCR (Figure 3A) and Western blot (Figure 3B) assays (Figure 3A,B). We performed colony formation, transwell, CCK8, and wound-healing assays to assess the effect of GATA3 on growth and migration in TNBC cells. The result in (Figure 3C,D) indicated that cell growth was reduced in GATA3-overexpressed cells, while GATA3 knocked down cells showed a high growth rate compared to the control cells. Compared to the control cells, the migration was restrained in cells with forced expression of GATA3, while faster migration was observed in the GATA3 knocked-down cells (Figure 3E and Appendix A). EMT and growth-related genes were investigated by Western blot and the result indicated that the overexpression of GATA3 increased E-cadherin expression, while the expression of Vimentin and β-catenin were decreased in the TNBC cells (Figure 3F). These findings suggested that GATA3 inhibited the expression of factors involved in EMT and growth. In addition, Notch signaling activity was examined by analyzing the expression of HES1 and HEY1 by RT-qPCR, their expression was reduced upon GATA3 overexpression (Figure 3G). These data demonstrated that GATA3 negatively regulated the TNBC malignancy by suppressing the cell migration and growth, as well as Notch signaling activation.

### 3.4. GATA3 Negatively Regulated MFNG Transcription by Abolishing Its Regulatory Activity in TNBC

The above findings illustrated that both MFNG and GATA3 affected Notch signaling activity, implying that there may be a correlation between MFNG and GATA3. GATA3 is a known transcription factor that controls the expression of several genes in the breast cancer [22,23,24]; therefore, we presumed that GATA3 could regulate the transcription of MFNG. To test this hypothesis, we performed an in silico analysis using the GEPIA2 database, the result implied that GATA3 expression was negatively correlated with MFNG expression (Figure 4A). In the clinical breast cancer tissues, IHC staining revealed a consistent expressing pattern in TNBC and statistical analysis showed a negative correlation between GATA3 and MFNG expression in TNBC (Figure 4B). Meanwhile, RT-qPCR and Western blot results indicated that the overexpression of GATA3 reduced the expression of MFNG at mRNA and protein levels in TNBC cells (Figure 4C,D). These results confirmed the negative correlation between GATA3 and MFNG suggesting that GATA3 might regulate the transcription of MFNG in TNBC cells. Furthermore, we analyzed the MFNG promoter sequence and found a potential GATA3 binding site (A[GATA]G) at the −358/−363 region. The wild-type reporter (−510 to +100 bp at MFNG promoter region) and mutant reporter (missing the binding site of GATA3) were created to examine the effect of GATA3 on the activity of the MFNG promoter (Figure 4E). The luciferase assay revealed that GATA3 significantly inhibited the activity of the wild-type MFNG promoter in the TNBC cells, but the mutant promoter activity basically remained unchanged (Figure 4F). This result demonstrated that GATA3 mediated the transcriptional inhibition of MFNG. In the meantime, the ChIP assay indicated that GATA3 remarkably enriched the promoter of MFNG (Figure 4G). These findings clarified that GATA3 was directly bound to the promoter of MFNG and suppressed its transcription.

### 3.5. miR205-5p Reduced the MFNG mRNA Level by Directly Binding to Its 3′ UTR Region and Inhibited the Malignancy of TNBC Cells

The lack of miR-205 in TNBC promotes cancer progression. It is indeed downregulated in TNBC and has a decreasing expression in metastatic breast tumors [25]. Since the onco-suppressive role of miR-205 in TNBC through targeting oncoproteins is evident [26,27], we predicted the target genes of miR205-5p by Targetscan and miRDB database analysis. Interestingly, we found that MFNG was a potential target of miR205-5p based on putative target sequences at the position 812–819 of the MFNG mRNA 3’ UTR (Figure 5A). To investigate the regulatory mechanism of miR205-5p on MFNG, we ectopically expressed miR205-5p mimic in TNBC cells and observed that the mimic significantly reduced the mRNA and protein levels of MFNG (Figure 5B,C). We created the reporters with the wild-type or mutant MFNG mRNA 3’ UTR region and performed a luciferase assay to verify this inhibitory activity. The result showed that miR205-5p was bound to the wild-type 3’ UTR region of MFNG mRNA, but not to the mutant 3′ UTR (Figure 5D). These results confirmed that miR205-5p abrogated the MFNG transcription and suppressed the malignancy of TNBC cells.

### 3.6. GATA3 Activated miR205-5p Transcription and Formed a Feed-Forward Loop with miR205-5p and MFNG in TNBC

Since GATA3 regulates microRNA expression in the breast cancer [28], we sought to determine whether GATA3 could control miR205-5p expression to cooperatively co-suppress MFNG expression. Therefore, we examined the expression of miR205-5p in GATA3-overexpressed TNBC cells and found that miR205-5p was upregulated (Figure 6A). Using the bioinformatics tool mentioned in the materials and methods section, we predicted that the miR205-5p promoter sequence harbored three putative GATA3 binding sites (Figure 6B). To confirm that miR205-5p was a potential target of GATA3, we constructed a wildtype reporter, mutant reporters of each GATA3 binding site (WT-Mut-S1, WT-Mut-S2, WT-Mut-S3), as well as combined mutant GATA3 binding sites (WT-Mut-Combined). Luciferase assay was performed in TNBC cells and found that increased luciferase activity of WT-GATA3 reporter was observed when overexpressing GATA3, while decreased luciferase activity of each mutant site appeared compared to the WT-GATA3. Moreover, GATA3 did not activate the luciferase activity of the WT-Mut-Combined reporter compared to the control (Figure 6C). Furthermore, the ChIP assay revealed a considerable enrichment of GATA3 on the miR205-5p promoter (Figure 6D). To further validate that miR205-5p was downstream of GATA3, we examined whether miR205-5p could restore the phenotype of GATA3 in TNBC. As indicated in (Figure 6E), compared to the control cells, miR205-5p mimics inhibited MFNG expression despite the absence of GATA3. Additionally, miR205-5p mimics rescued GATA3 inhibitory phenotype regarding the cell growth and migration in GATA3 knockdown cells by colony formation and transwell assays (Figure 6F,G). To further confirm the above results, we inhibited miR205-5p in GATA3-overexpressed cells and checked the expression of MFNG. The results were consistent with Figure 6E (Figure 6H). In general, our results discovered that GATA3 was a direct modulator of miR205-5p transcription and that miR205-5p mimic rescued the phenotype of GATA3 by the inhibition of MFNG expression, suggesting that GATA3/miR205-5p/MFNG form a novel feed-forward loop in the regulation of TNBC malignancy.

### 3.7. miR205-5p Exerted Effective Inhibition on The Malignancy of TNBC Cells

Black phosphorus has a unique advantage for drug delivery, to evaluate the tumor-suppressive role of miR205-5p in TNBC, we linked the synthetic miR205-5p with polyetherimide-black phosphorus (PEI-BP) nanoparticles that could accumulate in breast cancer and successfully obtained the composite nanoparticles (PEI-BP-miR205-5p) (Figure 7A–C). Thereafter, we investigated the effect of composite nanoparticles on TNBC cells in vitro and in vivo. We discovered that miR205-5p-PEI-BP exerted remarkable inhibition in cell growth of TNBC when compared to the control (Figure 7D). Meanwhile, the in vitro migration ability of TNBC cells was also significantly decreased by miR205-5p-PEI-BP (Figure 7E,F). To further confirm the inhibitory effect on TNBC cells, the tumorigenesis of TNBC cells was performed. Compared to the control, reduced growth and weight of tumors were obviously observed under the treatment of the composite nanoparticles (Figure 7G–I). Moreover, we checked the effect of miR-205-p application in the xenografts by analyzing the expression of MFNG in tumors. The results showed a dramatic decrease in protein and mRNA levels of MFNG expression in the tumors treated with miR205-5p-PEI-BP compared to the control (Figure 7J,K). These findings validated our in vitro observations, highlighting the oncosuppressive function of miR205-5p in TNBC by inhibiting MFNG expression. In particular, our findings indicated that miR205-5p had potential in TNBC therapy and provided a novel strategy for clinical TNBC treatment.

## 4. Discussion

TNBC is the most aggressive with the lowest survival rate among molecular subtypes of breast cancer [3]. Despite significant advances in therapeutic options for other subtypes of BC, the diagnosis and treatment for TNBC remain a significant challenge due to the lack of definitive targets. Thus, it is vital to identify promising therapeutic targets to improve the prognosis of TNBC.

The Notch signaling pathway plays a crucial role in the TNBC initiation and development [29]. As a key modulator of Notch signaling, MFNG plays dual functions in carcinogenesis as it can function as a tumor suppressor or an oncogene depending on the type of cancer [11,13]. The expression of MFNG negatively regulates JAG-1 resulting in the loss of tumorigenic phenotype and progression in the Human Papilloma Virus (HPV)-associated cervical neoplasia [30]. In lung cancer, MFNG suppresses Notch3 activation, subsequently inhibiting tumor growth in vitro and in vivo [11]. While our previous work demonstrated the oncogenic role of MFNG in claudin-low breast cancer by modulating the Pik3cg-driven Notch signaling [13], the precise mechanism as to how MFNG can be regulated in TNBC has not been reported elsewhere. In this study, we show that the increased expression of MFNG is associated with a more aggressive phenotype of TNBC. Furthermore, our data clearly demonstrated that transcription of MFNG could be prevented by GATA3 and miR205-5p, leading to the abrogation of Notch activity.

Similar to the other six members of GATA transcription factors, GATA3 possesses a zinc-finger DNA binding domain that binds to the 5′-(A/T) GATA (A/G)-3′ consensus domain [31]. Since it is crucial for breast tissue differentiation, the loss of GATA3 was strongly linked to the poor prognosis of TNBC [32]. Recently, the implication of GATA3 in breast cancer has emerged. Yan et al. reported the overexpression of GATA3 in MDA-MB-231 cells’ upregulated E-cadherin while inhibiting Vimentin expression, which reversed epithelial–mesenchymal translation that decreased cancer metastasis in vitro and in vivo [33]. Consistently, our findings showed a positive correlation between GATA3 and E-cadherin, a tumor suppressor associated with cell to cell adhesion which is an important event that contributes to tumor inhibition. Additionally, the overexpression of GATA3 in BT-549 and MDA-MB-231 cells significantly reduced cell migration in vitro and impaired mesenchymal phenotypes, while the opposite results were found following GATA3 knockdown. Our in vitro findings are consistent with the previous reports that GATA3 negatively regulates TNBC progression [20,34,35]. To further clarify how GATA3 diminishes TNBC aggressiveness, we demonstrated here that MFNG is a direct target of GATA3 and that loss of MFNG is associated with less invasiveness and TNBC development in vitro. In agreement with a number of studies [36,37], our observations revealed that the low expression of GATA3 is strongly associated with poor prognosis and showed that GATA3 is a strong and independent regulator of MFNG expression in TNBC, suggesting its importance as a powerful predictor of better clinical outcomes in TNBC patients. Our study underlines additional evidence supporting the crucial function of GATA3 as a potential tool for TNBC therapy.

Numerous studies revealed the tumor-suppressive roles of miRNAs in breast cancer through post-transcriptional regulation of their target genes. MiR-139 upregulation repressed breast cancer stem cells with mesenchymal characteristics and reduced cell invasiveness by downregulating the CXCR4/p-Akt axis [38]. Additionally, resistin was reported to dramatically decrease the levels of Let-7 miRNAs expression. The rescue of Let-7a inhibited the resistin-mediated tumorigenicity in the breast cancer [39]. In TNBC, miR205-5p functions as a tumor suppressor and predominantly targets oncogenic genes, including CLDN11 [40], HOXD9 [41], and HMBG [42]. Therefore, we performed several in vitro studies to investigate the possible relationship between miR205-5p and the oncogene MFNG in TNBC progression. Indeed miR205-5p was demonstrated to bind the 3′UTR region of MFNG and inhibit its expression followed by a remarkable decrease in cell proliferation and migration. Nanotechnology is becoming more prevalent in medical research and development and has recently been shown to be promising for cancer treatment. Nanoparticles, in particular, and their route of administration, have an impact on the efficacy of precision nanomedicine. Current cancer therapies have significant limitations due to localized drug delivery and therapeutic agent characteristics. In response to these challenges, nanoparticles have emerged as versatile drug vehicles. Furthermore, studies have shown that they can selectively target cancer cells, slow treatment resistance, and minimize side effects [43,44]. In this regard, black phosphorus (BP) has also been emerging as a suitable nanomaterial that allows drug delivery and therapeutics in cancers [45]. Our in vivo findings underlined the miR205-5P-PEI-BP had an effective inhibition on tumorigenesis of TNBC cells and revealed the crucial therapeutic role of miR205-5p in TNBC and BP was a superior nonmaterial in delivery drug. The previous evidence alongside our current data suggests that the miR205-5p/MFNG pathway may serve as a therapeutic target for breast cancer therapy and that the combination of miR205-5p and BP nanoparticles could serve as a potential approach in precision medicine to treat TNBC patients.

Transcriptional regulatory networks (TRNs) are important in oncogenic pathways discovery and gene regulatory systems. Feed-forward loops are essential motifs in TRNs and they play a synergistic role in several biological processes. Misregulation of feed-forward loops could lead to different diseases, including cancers [46]. Transcription factors (TFs) and microRNAs (miRNAs) are two kinds of gene expression regulators: TFs control gene expression by binding to promoter regions, whereas miRNAs control gene expression at the post-transcriptional level by attaching to 3′ untranslated regions. Importantly, transcription factors and microRNAs collaborate to coregulate the same target gene in the feed-forward loop system [47,48]. GATA3 was reported to upregulate the expression of miR-29b, leading to tumor microenvironment regulation and metastasis abrogation [27]. P53 also enhanced miR-205 activation to restrain cell proliferation in TNBC by targeting E2F1 [49]. Thus, we postulated that GATA3 could be a scaffold enhancer that modulates miR205-5p activation in TNBC. Our findings showed that GATA3 can bind to the miR205-5p promoter. Consequently, an increase in miR205-5p expression, which targets and abrogates the expression of MFNG, followed by TNBC slow migration and aggressiveness. The expression of miR205-5p is relatively low in the TNBC subtype [50], which could be one of the possible reasons that fuel TNBC aggressiveness. In addition, GATA3 mutations are frequently observed in several breast cancer clinical cases and most of those mutations are associated with GATA3 loss of function [51,52,53]. Thus, it could explain why miR205-5p is downregulated in TNBC while MFNG is subsequently upregulated.

## 5. Conclusions

Our study found that the high expression of MFNG in TNBC was associated with poor prognosis and promoted the malignant growth and migration of TNBC cells by activating Notch signaling. Meanwhile, our results demonstrated that GATA3 not only directly inhibited the transcription of MFNG through binding to its promoter but also suppressed the MFNG mRNA expression level by activating the transcription of miR205-5p which could bind to the 3′ UTR region of MFNG mRNA. These findings revealed the underlying mechanism of GATA3-miR205-5p on the inhibition of TNBC malignancy that GATA3/miR205-5p/MFNG formed a feed-forward loop (Figure 7L). In addition, miR205-5p could remarkably reduce the tumorigenesis of TNBC which lacks GATA3 expression. Therefore, our study suggests a novel feed-forward loop for the regulation of TNBC malignancy and provided a potential way of TNBC treatment.

## Figures and Tables

**Figure 1 cancers-14-03057-f001:**
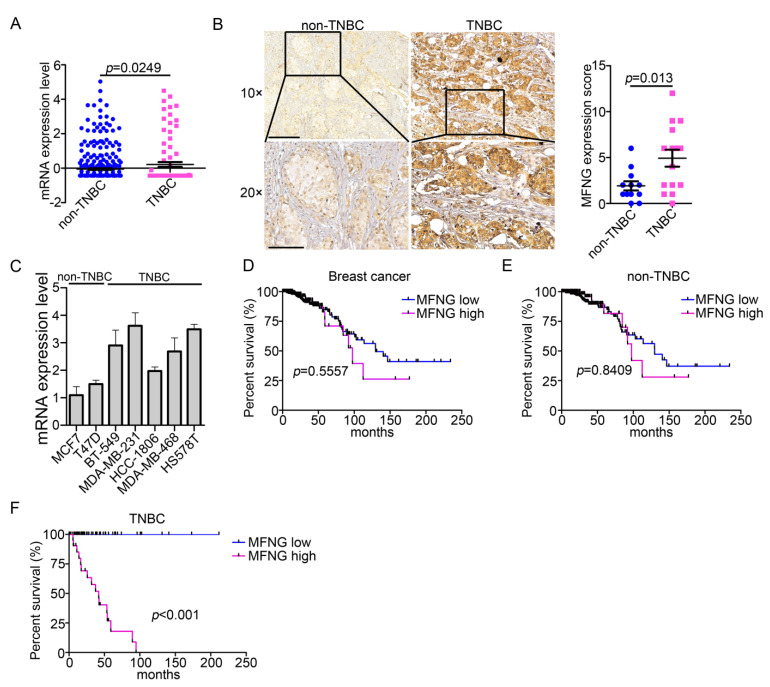
MFNG expression was significantly increased in TNBC tissues and positively associated with the poor prognosis of TNBC patients. (**A**) Based on the TCGA data (nature, 2012), we investigated the MFNG expression in TNBC (*n* = 389) and non-TNBC (*n* = 90) breast cancer samples. (**B**) Immunohistochemistry assay was used to assess the MFNG protein level in TNBC (*n* = 15) and non-TNBC (*n* = 12) breast cancer tissues (**left**), and the result of statistical analysis was shown (**right**). (**C**) MFNG mRNA expression level was examined in TNBC and non-TNBC cells in RT-qPCR, data representing the mean ± SD of three replicates. According to the MFNG expression level (Z-score) in breast cancers samples (TCGA, nature, 2012), patients were divided into MFNG low (Z-score ≤ −0.4319) and high (Z-score > −0.4319) expression groups. We analyzed the overall survival of breast cancers (**D**), non-TNBC (**E**), and TNBC (**F**) samples by using the Kaplan–Meier method, breast cancer (MFNG low, *n* = 358; MFNG high, *n* = 121; *p* = 0.5557), non-TNBC (MFNG low, *n* = 292; MFNG high, *n* = 99; *p* = 0.8409), and TNBC (MFNG low, *n* = 68; MFNG high, *n* = 23; *p* < 0.001). Survival curves were plotted by the Kaplan–Meier method and analyzed by the log-rank test. Statistical analyses were analyzed using a Student’s *t*-test.

**Figure 2 cancers-14-03057-f002:**
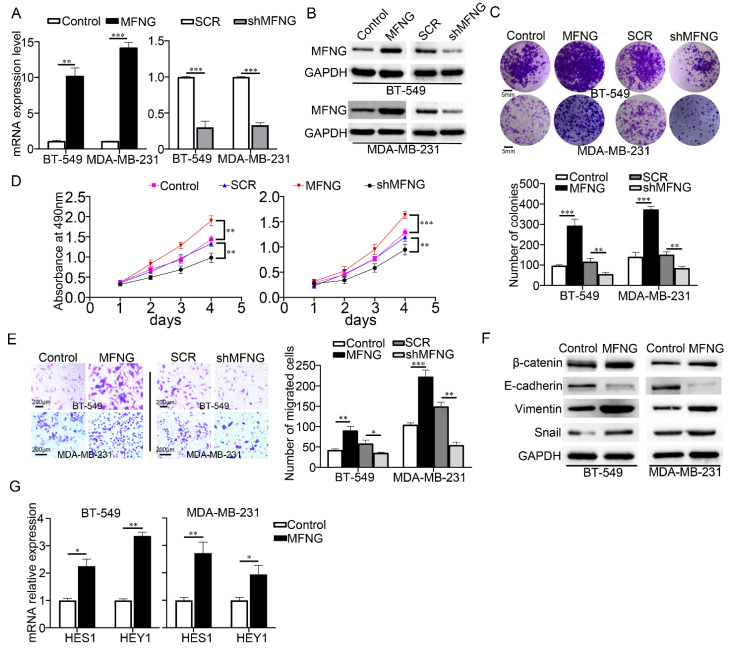
MFNG expression enhanced the cell growth and migration of TNBC cells. MFNG overexpression and knockdown were detected by RT-qPCR (**A**) and Western blot (**B**) in TNBC cells, SCR was short for Scramble and GAPDH served as an internal control. (**C**,**D**) The effect of MFNG on cell growth was examined in TNBC cells by colony formation and CCK8 assays. (**E**) The cell migration was assessed by transwell in MFNG-overexpressing or knockdown TNBC cells. (**F**) Western blot analysis of epithelial–mesenchymal transition (EMT) and growth-related genes in TNBC cells overexpressing MFNG, GAPDH was loaded as an internal control. (**G**) RT-qPCR analyzed the expression of Notch target genes HES1 and HEY1 in TNBC cells overexpressing MFNG, GAPDH served as an internal control. Data were analyzed using a Student’s *t*-test. All * *p* < 0.05, ** *p* < 0.01, *** *p* < 0.001.

**Figure 3 cancers-14-03057-f003:**
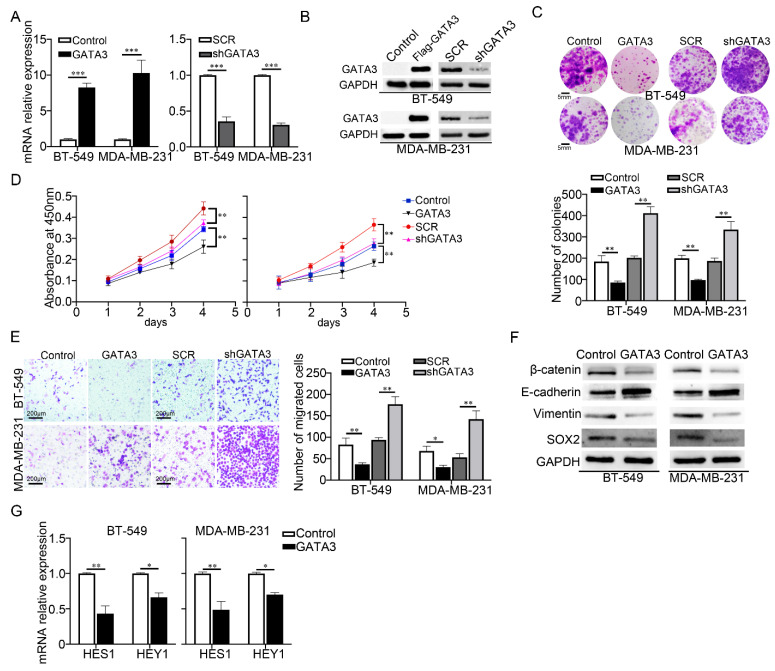
Decreased GATA3 elevated the malignancy of TNBC cells and promoted the activity of Notch signaling. The mRNA (**A**) and protein (**B**) expression levels of GATA3 were investigated in TNBC cells stably overexpressing or knockdown GATA3, SCR was short for Scramble and GAPDH served as an internal control. (**C**) Colony formation assay showing the number of colonies formed after transfecting indicated TNBC cells overexpressing or knockdown GATA3. (**D**) CCK-8 assay was conducted to evaluate the effect of GATA3 on cell growth in TNBC cells. (**E**) The role of GATA3 in TNBC cell migration was assessed by transwell migration assay in TNBC cells overexpressing or knockdown GATA3. (**F**) Western blot analysis of EMT and growth-related markers protein levels in the indicated TNBC cells, GAPDH was loaded as an internal control. (**G**) RT-qPCR analysis of the mRNA expression level of Notch target genes HES1 and HEY1 in GATA3-transfected TNBC cells, GAPDH served as an internal control. Data were analyzed using a Student’s *t*-test. All * *p* < 0.05, ** *p* < 0.01, *** *p* < 0.001.

**Figure 4 cancers-14-03057-f004:**
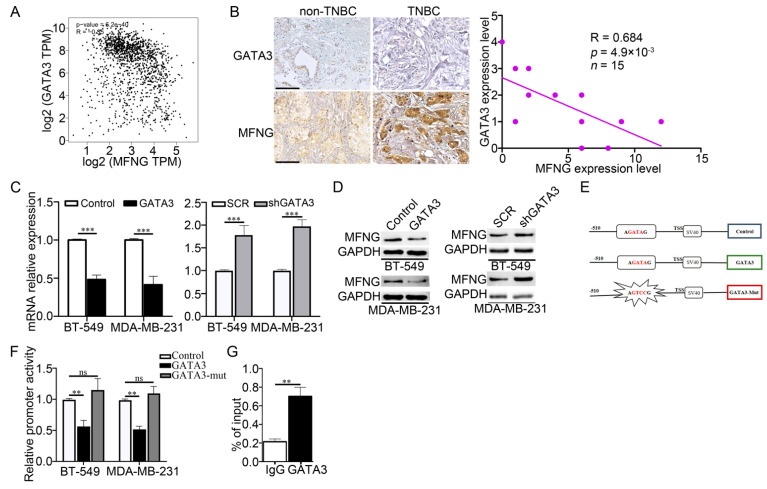
GATA3 negatively regulated MFNG transcription by abolishing its regulatory activity in TNBC. (**A**) Correlation analysis (GEPIA database) and IHC staining (**B**) revealed a negative correlation between the expression of GATA3 and MFNG in breast cancer, and statistical analysis in TNBC was shown, scale bar = 200 μm. (**C**) RT-qPCR and (**D**) Western blot analysis confirmed that MFNG was downregulated in GATA3 overexpressing or knockdown TNBC cells, SCR was short for Scramble and GAPDH served as an internal control. (**E**) The constructed vectors with putative GATA3 binding site in the MFNG promoter region and the mutated binding site with their corresponding control. The luciferase (**F**) and ChIP (**G**) assays were performed to estimate the regulation of GATA3 on MFNG transcription in MDA-MB-231 cells. Data were analyzed using a Student’s *t*-test. All ** *p* < 0.01, *** *p* < 0.001, ns: not significant.

**Figure 5 cancers-14-03057-f005:**
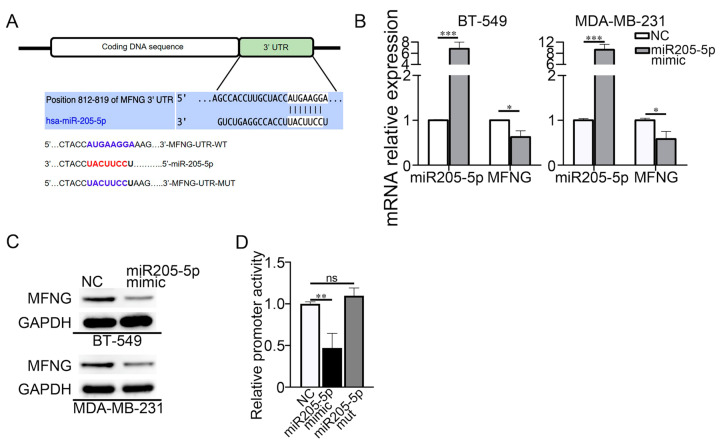
miR205-5p reduced the MFNG mRNA level by directly binding to its 3′ UTR region and inhibited the malignancy of TNBC cells. (**A**) The predicted binding sites of miR205-5p in the 3′-UTR of MFNG were performed using bioinformatics. (**B**,**C**) Overexpression of miR205-5p mimic in TNBC cells reduced the mRNA and protein expression levels of MFNG, NC was short for negative control and GAPDH was loaded as an internal control. (**D**) Ectopic expression of miR205-5p reduced the luciferase activity of wild-type 3′-UTR of MFNG in MDA-MB-231cells. Data were analyzed using a Student’s *t*-test. All * *p* < 0.05, ** *p* < 0.01, *** *p* < 0.001, ns: not significant.

**Figure 6 cancers-14-03057-f006:**
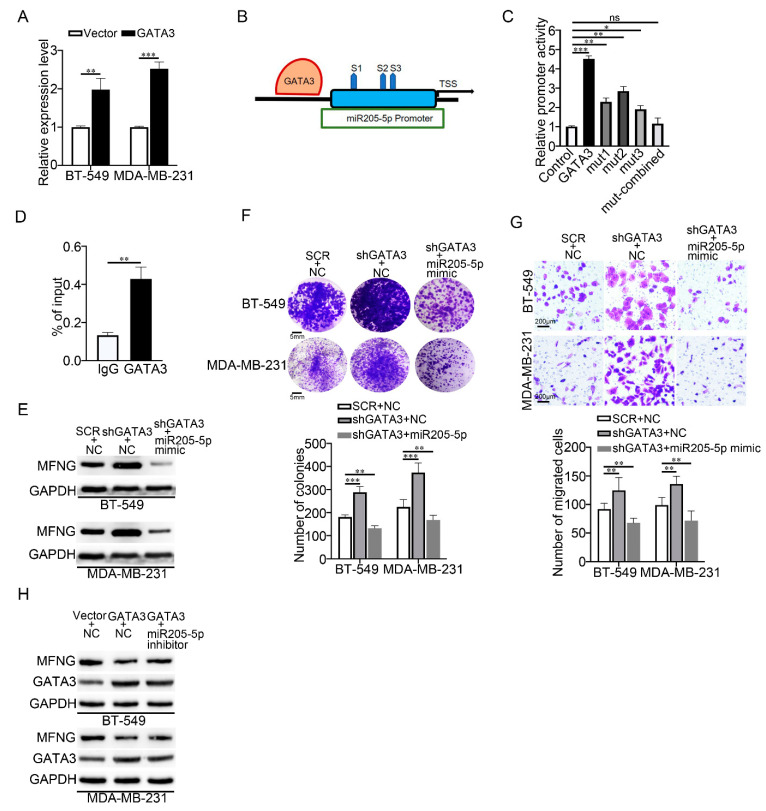
GATA3 activated miR205-5p transcription and formed a feed-forward loop with miR205-5p and MFNG in TNBC. (**A**) RT-qPCR analysis for miR205-5p in TNBC cells overexpressing GATA3. (**B**) The predicted binding sites for GATA3 on the miR205-5p promoter using bioinformatics. (**C**) The luciferase and (**D**) ChIP assays were performed to examine the regulation of GATA3 on miR205-5p transcription in MDA-MB-231 cells, IgG served as the negative control. (**E**) Protein expression level of MFNG in control, shGATA3/NC, and shGATA3/miR205-5p cells, GAPDH was loaded as the internal control. (**F**,**G**) Colony formation and transwell assays were used to assess the effect of miR205-5p mimic on cell growth and metastasis of GATA3-knockdown TNBC cells. (**H**) Protein expression level of MFNG in the control, GATA3/NC, and GATA3/miR205-5p inhibitor cells, GAPDH was loaded as the internal control. Data were analyzed using a Student’s *t*-test. All * *p* < 0.05, ** *p* < 0.01, *** *p* < 0.001, ns: not significant.

**Figure 7 cancers-14-03057-f007:**
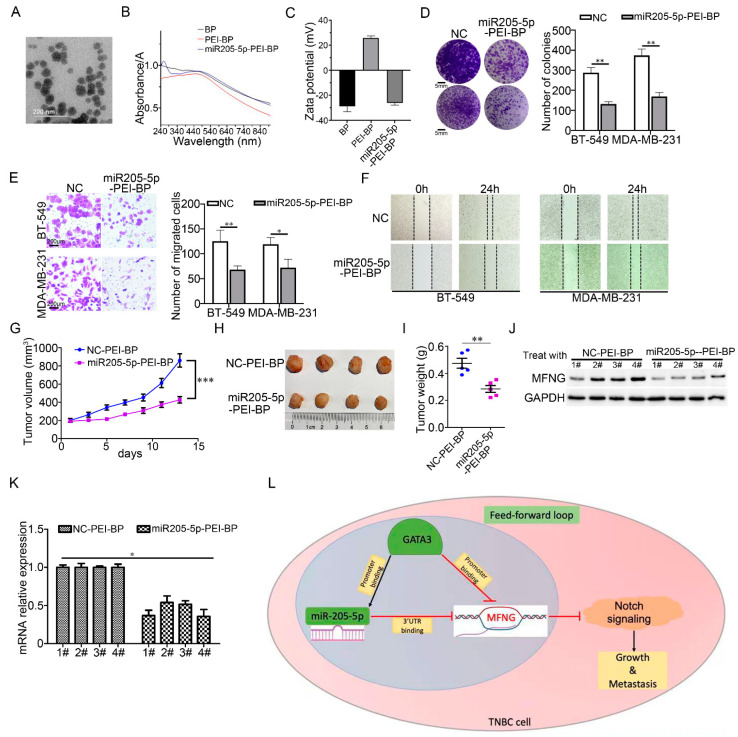
miR205-5p had potential in TNBC therapy. (**A**) Picture of miR205-5p-PEI-BP taken by transmission electron microscope, scale bar is 200 nm. (**B**) The light absorption spectrum of miR205-5p-PEI-BP. (**C**) The zeta electric potential of miR205-5p-PEI-BP. (**D**–**F**) The effect of miR205-5p-PEI-BP on cell growth and metastasis was examined by clone formation, transwell, and wound-healing assays (5 nmol), NC was the negative control of miR205-5p. (**G**) A time course of tumor growth. The decrease in tumor volume of the mice treated with miR205-5p-PEI-BP compared to their negative controls. Mice were treated with NC-PEI-BP (5 nmol) or miR205-5p-PEI-PB (5 nmol) twice a week for two weeks. (**H**,**I**) Following the sacrifice of mice, tumors were removed, photographed, and tumor weight was measured. (**J**) Western blot and (**K**) RT-qPCR analysis confirmed that MFNG was downregulated in the tumors treated with miR205-5p-PEI-PB, and GAPDH served as an internal control. (**L**) A comprehensive summary of the GATA3-miR205-5p feed-forward loop mechanism that targets MFNG and inhibits tumor growth and metastasis in TNBC. Data were analyzed using a Student’s *t*-test. All * *p* < 0.05, ** *p* < 0.01, *** *p* < 0.001.

## Data Availability

The authors confirm that the data supporting the findings of this study are available within the article and Appendix A. Clinical data obtained from TCGA (nature 2012) were also provided in the Appendix A.

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
