# Peer review of "Upregulated GATA3/miR205-5p Axis Inhibits MFNG Transcription and Reduces the Malignancy of Triple-Negative Breast Cancer"

_cancers, 2022, doi:10.3390/cancers14133057_

Round 1

Reviewer 1 Report

The authors investigated the regulatory mechanism of MFNG- They proposed a regulatory circuit involved GATA3/miR-205/MFNG in TNBC,

Specific points to be addressed:

  • A previous study (https://doi.org/10.1186/s12859-016-1196-1) found mir-205 to be associated with BC luminal subtype. In your opinion, how can explain it?
  • figure 1 A. there is not a great difference between TBNC and no TBNC. Could the author insert the p-value (not only pvalue<0.05)
  • figure 1 c. the authors should insert the p-value where significant
  • survival analysis. The authors should specify how they divided the samples into MFNG high and low. Median? Mean? Or other?
  • Line 270. Figure 2B should be a western blot. But in the main text the authors indicate that figure 2B is MFNG knockdown. Please check.
  • Figure 2A presents MFNG overexpression and MFNG knocked. What is the difference between control used in MFNG overexpression and scramble (SCR ?) used in MFNG knocked ? why did the author use different strategies?
  • The authors don’t cite western blot (figure 2B??) in the main text
  • I think the authors should adjust figure 2
  • I think control and scramble should be analyzed in the same experiment.
  • Same errors of Figure 3 A and B like figure 2 A and B.
  • Figure 4A. Is it obtained considering TBNC or all breast cancer?

Author Response

The authors investigated the regulatory mechanism of MFNG- They proposed a regulatory circuit involved GATA3/miR-205/MFNG in TNBC,

We thank you for your valuable comments and the time you devoted to revising our manuscript.

Specific points to be addressed:

  1. A previous study (https://doi.org/10.1186/s12859-016-1196-1) found mir-205 to be associated with BC luminal subtype. In your opinion, how can explain it?

Response: Thank you for this comment. In our view, the recommended article shows the significant role of miR-205 in controlling several crucial signaling pathways including HIF1 signaling, that Hsa-miR-205 is an oncosuppressive miRNA lost in BC and that Hsa-miR-205 has a potential therapeutic role by acting as biomarkers of the response to trastuzumab, and to neoadjuvant chemotherapy. We find this article valuable to upgrade our findings to highlight the vital function of miR-205 in breast cancer therapy. Therefore, we decided to cite this article (reference 26) as we believe it is useful to strengthen our hypothesis regarding miR-205 as a tumor suppressor in TNBC. The modified content was described in the 3.5 section and highlighted in yellow.

  1. figure 1 A. there is not a great difference between TBNC and no TBNC. Could the author insert the p-value (not only pvalue<0.05)

Response: We appreciate this valuable suggestion. In the revised manuscript we have inserted the actual p-value (P=0.0249), please see figure 1A.

  1. figure 1 C. the authors should insert the p-value where significant

Response: We thank you for the comment. Figure 1C was provided as a supplement to the clinical analysis (figure 1A) and tissue samples (1B) to validate that MFNG is highly expressed in TNBC compared to non-TNBC. Moreover, the data from non-TNBC was not enough to do a two-way ANOVA analysis that could show the significance between the two groups. Although we were unable to provide the p-value, the MFNG expression in TNBC cells was higher than that observed in non-TNBC.

  1. survival analysis. The authors should specify how they divided the samples into MFNG high and low. Median? Mean? Or other?

Response: We highly appreciate the comment and we agree that we should specify how we divided the samples into MFNG high and low. In the revised manuscript, we clarified that the patients in figure 1D-F were divided into MFNG low (Z-score≤-0.4319) and high (Z-score>-0.4319) expression groups according to the Z-score, and the correlation between MFNG expression and prognosis of patients was assessed, please see in figure 1 legend.

  1. Line 270. Figure 2B should be a western blot. But in the main text the authors indicate that figure 2B is MFNG knockdown. Please check.

Response: Thank you for this comment. Figure 2A-B shows the overexpression of MFNG in TNBC cell lines at both mRNA and protein levels. However, the sentence corresponding to these results in the old version of the manuscript sounded a bit confusing which is why in the revised manuscript we made it clear by indicating that figure 2B is western blot results. The modified content was described in 3.2 section and highlighted in yellow.

  1. Figure 2A presents MFNG overexpression and MFNG knocked. What is the difference between control used in MFNG overexpression and scramble (SCR ?) used in MFNG knocked ? why did the author use different strategies?

Response: We appreciate this comment. It is true that the control used for MFNG overexpression and knockdown are different and used different strategies. For the MFNG ectopic expression, we use MFNG overexpressing vector (pLenti-CMV-MFNG-FLAG-GFP-Puro) and pLenti-CMV-FLAG-GFP-Puro as its negative control. Whereas PLKO.1 vector was used to clone non-targeting control shRNA (SCR) and shRNAs against MFNG used in the knockdown experiment. Therefore, the strategies are different because we used different plasmids which use different systems (overexpression and knockdown).

  1. The authors don’t cite western blot (figure 2B??) in the main text

Response: We apologize for the confusion that the previous version of the manuscript made, in the current version we clearly mentioned figure 2B in the 3.2 section and highlighted it in yellow.

  1. I think the authors should adjust figure 2

Response: We thank you for the suggestion. However, the fact that the vectors we used for overexpression and knockdown are completely different thereby using different strategies, we believe that figure 2 is presented in the right way.

  1. I think control and scramble should be analyzed in the same experiment. Same errors of Figure 3 A and B like figure 2 A and B.

Response: We appreciate this comment. Like the response provided above (5. a), similarly to MFNG, in the experiments conducted to assess the overexpression and knockdown of GATA3, we used different vectors which presented different working systems. Therefore, we did not analyze the control (for overexpression) and the scramble (for knockdown) in the same experiment. However, we have made it clear in the revised manuscript the meaning of figure 3A and figure 3B.

  1. Figure 4A. Is it obtained considering TBNC or all breast cancer?

Response: We used the GEPIA2 database to analyze the possible correlation between GATA3 and MFNG. GEPIA2 is an online tool for analyzing the transcriptional profiles of human cancers and normal tissues, by using the TCGA database and the Genotype‐Tissue Expression (GTEx) projects. However, this database does not differentiate breast cancer subtypes, therefore, we used all breast cancer to analyze the MFNG-GATA3 correlation.

Reviewer 2 Report

1: The authors used only two TNBC cell lines in the study instead of at three. They did not justify which the two cells were picked other than been TNBC cell lines. TNBC is a heterogeneous group.

2: The rationale made in line 297 for investigating whether GATA-3 in the studying is not strong. The authors should make stronger case while the choose to look at GATA-3 because GATA-3 is not the only tumor suppressor that is suppressed in TNBC. 

3: Also additonal/stronger rationale should be made for choosing miR-205-5p.

3: The significant of the result 4F and 4G was not strongly discussed in the Discussion. Does the loss of GATA-3 contribute to the elevation level of MFNG? This should be linked to the miR-205-5p.

4: GATA-3 seem to have the ability to regulate the expression of MFNG directly by binding to the promoter and through miR-205-5p. Does this mean that GATA-3 binding on the MFNG has a suppressive function in the expression of MFNG. The authors need to clear this. One way will be to knockdown miR-205-5p in GATA-3 over-expressed cells and check the level of MFNG to demonstrate that it is not miR-205-5p resulting from the transcriptional activities of GATA-3 that is causing the down regulation of MFNG. Is the suppressive GATA-3 function reported in any literature?

5: Overall the results and experimental designs are good, but the discussion can be strengthen. The authors should extrapolate their findings. What is the implication of these findings and how can the application that focused on targeting of MFNG or GATA-3 or miR-205-5p help. The authors should discuss how activation of GATA-3 may be through the use of agonist can help in the treatment of TNBC patients. The should also talk about miRNA therapy and nanoparticles such as PEI-BP can also improve miRNA delivery and treatment. 

Author Response

Before addressing the comments given, we take this moment to thank you for the time you allotted to revising our manuscript by providing constructive comments and suggestions which helped us to improve the manuscript.

  1. The authors used only two TNBC cell lines in the study instead of at three.

Response: We do agree that three or more cell lines could strengthen even more our in vitro findings. However, several reports have used only two cell lines to draw a conclusion on cellular experiments. We also believe that the two cell lines used in our study were enough to provide reliable in vitro data.

  1. They did not justify which the two cells were picked other than been TNBC cell lines. TNBC is a heterogeneous group.

Response: Thank you for this comment. We agree that a justification would be important to underline why we chose the two cell lines over others. Therefore, in the revised manuscript we highlighted the reason (This was described in 2.1 section and highlighted in yellow), and the following is a detailed explanation. It is true that TNBC is a heterogeneous group as it is subdivided into six subgroups: 2 basal-like (BL1or A and BL2 or B), an immunomodulatory (IM), a mesenchymal (M), a mesenchymal stem-like (MSL), and a luminal androgen receptor (LAR). The reason we selected BT-549 and MDA-MB-231 for further experiments, is that they share several features including the same subtype (basal 2). Besides, they are all epithelial, and invasive, and show the lowest expression of genes involved in epithelial cell-cell adhesion. They are also significantly enriched in EMT and stem cell-like features while showing a low expression of luminal and proliferation-associated genes (https://doi.org/10.1186/bcr2635). Since these two cell lines also lack the growth factor receptor HER2, they represent a good model of triple-negative breast cancer. Regarding the in vivo experiment, we chose the MDA-MB-231 cell line over others due to its special characteristics as reported in several studies. Additionally, some luminescent clones of MDA-MB-231 are available and well-characterized in vivo for good imaging (https://doi.org/10.1016/B978-0-12-415894-8.00040-3).

  1. The rationale made in line 297 for investigating whether GATA-3 in the studying is not strong. The authors should make stronger case while the choose to look at GATA-3 because GATA-3 is not the only tumor suppressor that is suppressed in TNBC.

Response: We appreciate your comment. In the revised manuscript, we added several points in the introduction part to underline the function of GATA3 as a master regulator of mammary gland differentiation and homeostasis (please see 3.3 section). We have also inserted an argument that strengthens the role of loss or mutations of GATA3 causing aggressiveness of breast cancer. In addition to GATA3 oncosuppressive bio-function, we used bioinformatics to analyze the promoter sequence which we found to have GATA3 binding sites, we then speculated that GATA3 could bind and control the expression of MFNG. Also, the fact that the overexpression of GATA3 reduced the Notch activity (Figure 3G), and because we already know that MFNG upregulates Notch, this result prompted us to think that GATA3 could have a significant role in MFNG regulation.

  1. Also additonal/stronger rationale should be made for choosing miR-205-5p.

Response: We thank you for this valuable comment. Now the results and the discussion parts of the revised manuscript emphasize the significance of miR-205-5p in the treatment of TNBC (please see 3.5 section and lines 567-573).

  1. The significant of the result 4F and 4G was not strongly discussed in the Discussion. Does the loss of GATA-3 contribute to the elevation level of MFNG? This should be linked to the miR-205-5p.

Response: Thank you for your comment. We apologize for any misunderstanding caused by Figures 4F and 4G. The two figures represent the luciferase and ChIP assays conducted to confirm the binding activity of GATA3 to the promoter of MFNG. It was explained in the results part (please see 3.4 section) and was also highlighted in the discussion part (lines 538-541).

  1. GATA-3 seem to have the ability to regulate the expression of MFNG directly by binding to the promoter and through miR-205-5p. Does this mean that GATA-3 binding on the MFNG has a suppressive function in the expression of MFNG. The authors need to clear this.

Response: We appreciate your comment. In our study, we showed the MFNG promoter has a potential binding site for GATA3 and demonstrated that GATA3 directly binds to the MFNG promoter and suppresses its transcription. To confirm this inhibitory activity of GATA3 on MFNG, we performed a luciferase assay to examine the GATA3 binding ability as shown in figure 4F, in which the overexpression of GATA3 significantly reduced the MFNG promoter activity. To reaffirm this result, we carried out a ChIP assay, please see figure 4G.

  1. One way will be to knockdown miR-205-5p in GATA-3 over-expressed cells and check the level of MFNG to demonstrate that it is not miR-205-5p resulting from the transcriptional activities of GATA-3 that is causing the down regulation of MFNG.

Response: We thank you for this valuable comment which we also believe is important to underline the function of GATA3 on the expression of MFNG. Even though we described in detail how GATA3 blocks MFNG expression, we do agree that adding this experiment will strengthen the tumor-suppressive of GATA3 in TNBC. Therefore, in the revised manuscript, we evaluated whether GATA3 affects MFNG expression through miR-205-5p. We inhibited miR-205-5p using hsa-miR-205-5p inhibitor sense 5`CAGACUCCGGUGGAAUGAAGGA3`. The results were consistent with what we found early in Figure 6E as shown in figure 6H of the revised manuscript. We demonstrated that MFNG is regulated through the feed-forward loop.

  1. Is the suppressive GATA-3 function reported in any literature?

Response: Yes, the suppressive function of GATA3 has previously been reported in several studies, please have a look at some of those reports: (1) Doi: 10.1016/j.ccr.2008.01.011; (2) Doi: 10.1038/ncb2672; (3) Doi: 10.1038/onc.2017.165; (4) Doi: 10.1186/s12935-020-01424-3; (5)Doi: 10.1074/jbc.M110.105262.

  1. Overall the results and experimental designs are good, but the discussion can be strengthen. The authors should extrapolate their findings. What is the implication of these findings and how can the application that focused on targeting of MFNG or GATA-3 or miR-205-5p help. The authors should discuss how activation of GATA-3 may be through the use of agonist can help in the treatment of TNBC patients. The should also talk about miRNA therapy and nanoparticles such as PEI-BP can also improve miRNA delivery and treatment.

Response: Thank you for your comment. In the revised manuscript, we further discussed our findings and underline how our study is an additional contribution to the current breast cancer research, particularly TNBC. Based on our findings, we also underlined the potential role of GATA3 in the TNBC treatment. Finally, we emphasized the application of nanotechnology in delivering suppressor miRNAs to tumors in a precise manner which is a novel and reliable tool for cancer therapy, please see lines 558-566.

Reviewer 3 Report

In this study the authors describe the prognostic impact of MFNG expression in triple-negative breast cancer and reveal an association with bad prognosis. Furthermore, they elucidate the underlying mechanisms and suggest GATA3 and miR-205-p as modulators of MFNG expression. Moreover, they show mIR-205-p regulation via GATA3. Finally the study provides insights into the therapeutic usability of miR-205-p by deploying black phosphorus as a vehicle. 

The overall scientific concept is clear and straight forward. The authors are adressing a severe clinical issue by providing deeper insights into TNBC progression and therapeutic strategies. All the experiments have been performed properly and the in vitro data consist of two different cell lines, both showing coherent results. 

There are just some minor comments from my side that might make it easier for the scientific community to follow the data aquisition. In particular in terms of methods reproducability.

  1. It would be good to include the actual antibody concentrations in the MM part (chapter 2.4).
  2. A brief definition of the non-TNBC cohort would help to understand de actual discrimination better. Moreover, which actual dataset was used fo the in silico analysis e.g. microarray data or RPPA (chapter 2.6).
  3. chapter 2.7 the actual cell count would be helpful. Here only a concentration is given.
  4. The evaluation of the migration assay was performed by staining the membrane or cells left in the bottom of the well ? (chapter 2.8)
  5. Which reporter system was used for the IHC ? (chapter 2.10)
  6. The in vivo intervention experiment states a negative control, it would be good to know if it was BP without miRNA or just solvent control ? (chapter 2.13).
  7. Especially for the mouse experiment it can be hard to state normal distribution for the use of T-tests. The effects look quite solid at all, but a cross check could be helpful.

There are some minor remarks on the results aswell.

  1. Why was the IHC staining for MFNG not quantified. It could further strengthen the TCGA data ? How was the cut-off defined for MFNG high and low ? (Fig 1).
  2. The authors provided pictures of the whole blots for all the analysis. However it looks like the single lanes and columns have been cut. Is there a particual reason for this ? It also seems that the marker is missing sometimes, which makes it hard to estimate the actual band size. Is there also a reason for this ? (Fig 2 -)
  3. You showed stainings of MFNG and GATA 3 examplarily. Would it be possible to show an actual correlation of your MFNG and GATA3 stainings. This would strengthen the TCGA data. (Fig 3).
  4. The effects of miR-205-p application in the xenograft model look very promising. Did you analyse the expression of MFNG in the tumors of the animals to validate the decreased expression here ? This would round up the hypothesis in vivo ?
  5. The application of miR-205-p in TNBC patients could be a potential strategy for further therapy. It could be a good addition to the discussion to suggest an indication for the therapy in the clinical workflow, eg which patients would be treated and maybe comment on the potential side effects of miR-205 increase.

Author Response

In this study the authors describe the prognostic impact of MFNG expression in triple-negative breast cancer and reveal an association with bad prognosis. Furthermore, they elucidate the underlying mechanisms and suggest GATA3 and miR-205-p as modulators of MFNG expression. Moreover, they show mIR-205-p regulation via GATA3. Finally the study provides insights into the therapeutic usability of miR-205-p by deploying black phosphorus as a vehicle.

The overall scientific concept is clear and straight forward. The authors are adressing a severe clinical issue by providing deeper insights into TNBC progression and therapeutic strategies. All the experiments have been performed properly and the in vitro data consist of two different cell lines, both showing coherent results.

Response: Thank you for your kind words. We appreciate the time you allotted to reviewing our manuscript.

There are just some minor comments from my side that might make it easier for the scientific community to follow the data aquisition. In particular in terms of methods reproducability.

  1. It would be good to include the actual antibody concentrations in the MM part (chapter 2.4).

Response: Thank you for the suggestion, in the revised manuscript we have added the actual antibody concentrations in the materials and methods part, section 2.4. Anti-MFNG antibody (NBP1-79288, 1:1000) was purchased from Novus Biologicals (USA), anti-E-Cadherin antibody (24E10, 1:100), anti-Flag antibody (D6W5B, 1:1000), anti-Vimentin antibody (D21H3, 1:1000), anti-SLUG antibody (C19G7, 1:1000), anti-SOX2 antibody (D9B8N, 1:1000), anti-ß-Catenin antibody (D10A8, 1:1000) were purchased from Cell Signaling Technology (Danvers, MA, USA), anti-GATA3 antibody (HG3-3, 1:1000) was purchased from Santa Cruz Biotechnology, Inc., and anti-GAPDH antibody (1E6D9, 1:10000) was purchased from Proteintech, IL (USA). The revised part was highlighted in yellow.

  1. A brief definition of the non-TNBC cohort would help to understand de actual discrimination better. Moreover, which actual dataset was used fo the in silico analysis e.g. microarray data or RPPA (chapter 2.6).

Response: We appreciate the comment. The 2.6 section in materials and methods part has been revised and describes clearly the “non-TNBC” group which was the breast cancers excluding TNBC presented ER, PR, and/or HER2 positive, the revised part was highlighted in yellow. For in silico analysis, we extracted the data under “mRNA/miRNA expression Z-scores (all genes)” in cBioPortal for Cancer Genomics database.

  1. chapter 2.7 the actual cell count would be helpful. Here only a concentration is given.

Response: Thank you for your suggestion. We apologized for the unnecessary typo and the confusion it brought. In the revised manuscript, we indicated the actual number of cells we used which were 3000 cells in each well of the 96-well plate, please see 2.7 section in materials and methods part and the revised part was highlighted by yellow.

  1. The evaluation of the migration assay was performed by staining the membrane or cells left in the bottom of the well? (chapter 2.8)

Response: We have made a clarification on the transwell migration assay. As stated in the revised manuscript (2.8 section in materials and methods part), “After 24 hours, the upper chamber cells were removed, and cells that had moved into the lower chamber were fixed with a 4% paraformaldehyde (PFA), and stained with crystal violet.” We stained the membrane (lower chamber) containing migrated cells, and the cells in the upper chamber were wiped with a wet cotton swab. After it was slightly dry, the chamber was placed under an inverted microscope, and photographs were captured in 5 random places. The average of each group was measured, and the change in cell migration ability was calculated by the number of cells passing through the membrane. The revised part was highlighted in yellow.

  1. Which reporter system was used for the IHC? (chapter 2.10)

Response: Thank you for the comments, in the 2.10 section of the revised manuscript we indicated the reporter system we used for the IHC which is “3,3'-diaminobenzidine (DAB)” which is a horseradish peroxidase (HRP) substrate that is used for secondary HRP-conjugated antibody detection to detect the antibody-bound antigen. Moreover, we added the detail of how we scored each immunohistochemistry section for analysis. The revised part was highlighted in yellow.

  1. The in vivo intervention experiment states a negative control, it would be good to know if it was BP without miRNA or just solvent control? (chapter 2.13).

Response: We thank you for the suggestion, in the 2.12 and 2.13 section of current manuscript we clearly mentioned that the negative control was NC-PEI-BP. When the tumors reached the palpable size, four of eight mice were injected with miR-205-5p-PEI-BP mimic whereas the remaining four were injected with NC-PEI-BP as control mice. The revised part was highlighted in yellow.

  1. Especially for the mouse experiment it can be hard to state normal distribution for the use of T-tests. The effects look quite solid at all, but a cross check could be helpful.

Response: We thank you for this comment. To analyze the tumor data, we used Two-way ANOVA analysis, which we have made clear in section 2.13 of the revised manuscript. The revised part was highlighted in yellow.

There are some minor remarks on the results aswell.

  1. Why was the IHC staining for MFNG not quantified. It could further strengthen the TCGA data ? How was the cut-off defined for MFNG high and low? (Fig 1).

Response: Thank you for the comment. In the revised manuscript, we have added a quantitative result of MFNG protein expression in Figure 1B, which indicated that MFNG protein expression was higher in TNBC tissues than in non-TNBC. How to score the MFNG protein expression was explained in Method (2.10 section in materials and methods part). The patients in Figure1 D-F were divided into MFNG low (Z-score ≤-0.4319) and high (Z-score >-0.4319) expression groups according to the Z-score which is indicated in figure 1 legends of the revised manuscript, and the correlation between MFNG expression and prognosis of patients was assessed.

  1. The authors provided pictures of the whole blots for all the analysis. However it looks like the single lanes and columns have been cut. Is there a particual reason for this?

Response: We thank you for the comment. The reason why some lanes have been cut is that we run only two samples on the whole gel (control and treated by overexpression or knockdown) for each cell line (BT-549 and MDA-MB-231), so because most of the lanes on the gel were empty (no samples loaded) we decided to cut that part and retain only the lanes with samples. The columns have been cut as well, this was due to the container we used to incubate the first antibodies, which was not big enough to hold the whole blot, so we decided to cut a part that was not containing the protein of interest. The reason we chose such a container is that it has several holes which could incubate several blots with different antibodies at the same time.

It also seems that the marker is missing sometimes, which makes it hard to estimate the actual band size. Is there also a reason for this? (Fig 2 -)

Response: Thank you for the comment. We agree that the marker is not clear at some blots in the white caption, the reason for this could be the caption error which we apologize for, but in the black caption, the marker is clear. In order to clarify this matter, here we provide the black caption of the blots together with the type of container we used to incubate the specific antibodies. 

  1. You showed stainings of MFNG and GATA 3 examplarily. Would it be possible to show an actual correlation of your MFNG and GATA3 stainings. This would strengthen the TCGA data. (Fig 3).

Response: Thank you for the suggestion, we have added the actual correlation between MFNG and GATA3 expression in TNBC (n =15), and the result of statistical analysis showed negative correlation between them (please see revised Fig 4B).

  1. The effects of miR-205-p application in the xenograft model look very promising. Did you analyse the expression of MFNG in the tumors of the animals to validate the decreased expression here? This would round up the hypothesis in vivo?

Response: We thank you for this very helpful comment. We agree that the analysis of MFNG expression in the animal tumors could validate the decrease in its expression and we believe that adding these results would uplift our findings. Therefore, we extracted mRNA and protein from the tumors to examine the expression of MFNG. Expectedly, the in vivo results were consistent with those found in vitro (please see revised Fig 7J and K). The revised contents were highlighted in 3.7 section.

  1. The application of miR-205-p in TNBC patients could be a potential strategy for further therapy. It could be a good addition to the discussion to suggest an indication for the therapy in the clinical workflow, eg which patients would be treated and maybe comment on the potential side effects of miR-205 increase.

Response: We deeply appreciate this valuable suggestion. We have emphasized the actual role of miR-205-5p clinical for cancer therapy highlighting the most likely group of people who could benefit from such therapy, please see lines 567-573. Based on our current data, we could not make any comment on the potential side effects of the miR-205-5p increase. Fortunately, our study is ongoing research in which we are planning to find several points including an investigation of whether MFNG expression could modulate intercellular communication between tumor cells and other cells in the tumor microenvironment. During our study, it will be a great opportunity to examine the potential side effects which may be associated with increased miR-205-5p as you suggested.